# Structure and Long-Term Stability of the Microbiome in Diverse Diatom Cultures

Marcelo Malisano Barreto Filho,[a] Melissa Walker,[a] Matt P. Ashworth,[b] J. Jeffrey Morris[a]

aDepartment of Biology, University of Alabama at Birmingham, Birmingham, Alabama, USA

bDepartment of Integrative Biology, University of Texas, Austin, Texas, USA

Marcelo Malisano Barreto Filho and Melissa Walker contributed equally to this work. Both are Ph.D. students; Marcelo Malisano Barreto Filho is listed first because this article will be a part of his Ph.D. dissertation.

**ABSTRACT** Microalgal cultures are often maintained in xenic conditions, i.e., with associated bacteria, and many studies indicate that these communities both are complex and have significant impacts on the physiology of the target photoautotroph. Here, we investigated the structure and stability of microbiomes associated with a diverse sampling of diatoms during long-term maintenance in serial batch culture. We found that, counter to our initial expectation, evenness diversity increased with time since cultivation, driven by a decrease in dominance by the most abundant taxa in each culture. We also found that the site from which and time at which a culture was initially collected had a stronger impact on microbiome structure than the diatom species; however, some bacterial taxa were commonly present in most cultures despite having widely geographically separated collection sites. Our results support the conclusion that stochastic initial conditions (i.e., the local microbial community at the collection site) are important for the long-term structure of these microbiomes, but deterministic forces such as negative frequency dependence and natural selection exerted by the diatom are also at work.

**IMPORTANCE** Natural microbial communities are extremely complex, with many more species coexisting in the same place than there are different resources to support them. Understanding the forces that allow this high level of diversity has been a central focus of ecological and evolutionary theory for many decades. Here, we used stock cultures of diatoms, which were maintained for years in continuous growth alongside populations of bacteria, as proxies for natural communities. We show that the bacterial communities remained relatively stable for years, and there is evidence that ecological forces worked to stabilize coexistence instead of favoring competition and exclusion. We also show evidence that, despite some important regional differences in bacterial communities, there was a globally present core microbiome potentially selected for in these diatom cultures. Understanding interactions between bacteria and diatoms is important both for basic ecological science and for practical science, such as industrial biofuel production.

**KEYWORDS** algal culture, black queen hypothesis, diatom, historical contingency, microbiome, negative frequency dependence, phycosphere

Microalgae and bacteria form the central axis of most aquatic communities, largely controlling the energy flux and biogeochemistry of aquatic ecosystems (1, 2). Primary production by microalgae provides energy and carbon for the bacteria as well as higher trophic levels, whereas bacteria remineralize necrotic microalgal biomass, yielding the necessary nutrients for continued growth (2–4). In addition to these basic interactions, both microalgae and bacteria also produce a wide variety of secondary

Address correspondence to J. Jeffrey Morris, evolve@uab.edu.

Diatom cultures harbor very diverse bacterial microbiomes that are stable for years

metabolites that are either actively secreted or passively leaked into the environment, creating a rich "marketplace" of molecules that can be exploited by any organism in the system (5–11). Both bacteria and microalgae also perform valuable ecosystem services, such as detoxification (12–14) and biofilm deposition (15), that potentially benefit their neighbors, regardless of species. Collectively, the concentration gradient of resources and services extending outward from the edge of microalgal cells is known as the phycosphere (16). The phycosphere is thought to be a driving structural force in open-water bacterioplanktonic ecosystems, especially in the vicinity of larger eukaryotic microalgae, such as diatoms (6).

Several approaches, such as atomic force microscopy and CARD-FISH (fluorescence *in situ* hybridization combined with catalyzed reporter deposition), have been employed to directly study interactions within the phycosphere (17, 18). However, the ephemeral nature of these communities and the ease with which they are disrupted during collection and examination make these efforts extremely technically challenging. An alternative approach has been to study microalga-bacterium interactions in cultures. These experiments typically involve bringing both microalgae and bacteria into clonal, axenic culture and then mixing them together, either in cocultures or in single-organism cultures treated with filtered supernatants from the proposed partner organism. These experiments have been invaluable for understanding the types of interactions that occur between algae and bacteria, including (i) bacterial production of hormones (7, 19, 20) and vitamins (8, 21) which influence algal morphology and physiology, (ii) bacterial removal of reactive oxygen species from the environment (12–14), (iii) allelopathy (22–24), (iv) bacterial recruitment of microalgae to benthic surfaces (25, 26), and (v) bacterial chemotaxis toward, and catabolism of, microalgal exudates (9, 11, 16). Unfortunately, like all pure-culture experiments, these studies are biased toward taxa that are readily grown axenically in the laboratory and may not reflect processes occurring in the actual ocean (27). In particular, the more obligate an interaction becomes, the more difficult it becomes to discover it through pure-culture experiments, as one or both partners may fail to grow robustly, or at all, as an axenic culture. An important example of this phenomenon is the inability of axenic cultures of the highly abundant open ocean cyanobacterium *Prochlorococcus* to tolerate even mild stresses experienced commonly in the laboratory (13, 14, 28). Anecdotally, we have found that many diatom cultures are difficult to render axenic, possibly due to dependencies similar to those observed in *Prochlorococcus*. Indeed, high-throughput methods have revealed that many or even most marine bacteria are similarly dependent on inputs from their community neighbors to grow in culture (29–31).

A classic strategy for studying difficult-to-cultivate microorganisms is the enrichment culture. Pioneered by Beijerinck in the 19th century (32), this method involves adding nutrients to a natural sample in order to study how the relatively intact community responds to experimental manipulations. In the past, enrichment cultures allowed researchers to discover a number of microbially mediated biogeochemical processes before they were able to isolate the responsible organisms into pure culture (33). Currently, the advent of metagenomic methods has led to a resurgent interest in enrichment methods for the study of microbial ecology (34). Here, we approached xenic laboratory cultures of diatoms as a proxy for natural phycosphere communities to characterize the short and long-term fluctuations in associated bacterial populations in order to better understand the nature of the relationships driving these interactions. The bacteria in these cultures probably originated from the natural communities from which they were taken and were able to grow because the diatoms served as the added "nutrients."

Diatoms are often isolated into "unialgal" cultures when single cells are manually picked from a mixed assemblage under magnification and placed in media supplemented with nitrogen, phosphorus, silicate, vitamins, and trace metal salts (35). Importantly, there is typically no added carbon in the medium, yet bacterial assemblages persist alongside diatoms even after years of maintenance as serial batch cultures. Photosynthetically produced carbon exudates from the diatoms themselves are possibly the sole source of

Microbiology
Spectrum

energy available to these bacterial communities. Reasoning that xenic diatom cultures essentially function as phycosphere enrichment cultures, we investigated the composition of the bacterial component of several marine diatom cultures representing both a phylogenetically and phenotypically broad sampling of diatom diversity as well as temporal cross sections of several cultures representing different lengths of time in culture, from less than 1 to almost 8 years. Remarkably, we found that these consortia remained diverse for years, and evenness diversity tended to increase over time. Collection site, and hence the initial bacterial inoculum, was more important than diatom species for structuring bacterial communities. Nevertheless, some bacterial taxa were commonly present in many cultures despite having been collected at different locations and times, suggesting the existence of a core diatom-associated microbiome.

## RESULTS

We sequenced the 16S rRNA V4 barcode from 15 bacterial communities, representing 10 unique cultures of 9 different diatom species (Table 1). Two araphid pennate cultures (*Astrosyne radiata* and *Florella pascuensis*) and one mediophycean centric culture (*Roundia cardiophora*) had multiple pellets sampled, spanning one or more years of culture maintenance postisolation. In the case of *A. radiata*, a second strain isolated from the same geographic location 6 years after the first strain was also pelleted and analyzed. Regarding geographic variation, four strains (representing araphid pennates and mediophycean and coscinodiscophycean centrics) were isolated from the same location—Gab Gab Beach, Guam—over a period of several years and analyzed. In total, we were able to compare the influence of sampling location, diatom species, and length of cultivation on the structure of the diatom-associated microbial consortia occupying these cultures.

Across all samples and after quality control and subsampling, we analyzed 927 operational taxonomic units (OTUs), with only 2 OTUs being present in all 15 communities and representing only 0.4% of total reads. On average, samples had 128.2 OTUs, with a minimum of 63 and a maximum of 298 (Table 2). Four hundred forty-one OTUs were found only as single copies in any given sample, and a further 184 were never more than 0.1% of any community. Other OTUs were very common, with 32 OTUs being found in more than half of the samples. Even if we restricted our analysis to these most ubiquitous OTUs, the least diverse sample, *Neosynedra provincialis*, had 18 distinct bacterial taxa collectively representing more than one-third of its detected sequences. Less conservatively, the Chao richness estimator suggested that the cultures had on average 252 OTUs, with a maximum of 678 and a minimum of 99. Thus, we conclude that all of our diatom cultures harbored diverse and surprisingly rich assemblages of bacteria, even after years in culture.

We next examined trends in richness and evenness diversity over time. Our initial expectation was that diversity would start out relatively high, reflecting the conditions in the surrounding seawater from which the diatoms were cultured, but would decline with time as superior competitors under the conditions of the laboratory excluded niche rivals. In fact, the opposite trend was observed, with evenness (but not richness) diversity significantly increasing over time. The best-fit model of the Shannon and inverse Simpson diversity statistics as a function of time (according to Bayesian information criterion analysis in comparison with simple linear and rectangular-hyperbola models) was a simple power-law model (exponential parameter $a = 0.147$ and $0.331$, respectively; $P < 0.001$) (Fig. 1A and B), whereas the Chao richness indicator was better fit by a linear model with slope not significantly different from 0 (Fig. 1D). The proportion of the community constituted by the most abundant OTU (Table 2; also, see Table S2 in the supplemental material) also decreased according to a power law over time ($a = -0.182$, $P < 0.001$) (Fig. 1C). Thus, the increased evenness diversity likely does not reflect the appearance of spurious contaminants or sequencing artifacts, but rather an evening-out of the community over time, with less intense overdominance by single taxa. Importantly, these trends in evenness diversity observed across all samples also

Microbiology Spectrum

**TABLE 1** Diatom microbiome samples analyzed[a]

| Sample[b] | Diatom species | Collection site | Collection date | Time (days)[c] |
|---|---|---|---|---|
| 1.1 | *Astrosyne radiata* | Gab Gab, Guam | June 2008 | 183 |
| 1.2 | | | | 330 |
| 1.3 | | | | 751 |
| 1.4 | | | | 2,727 |
| 2.1 | *Roundia cardiophora* | Achang, Guam | June 2008 | 347 |
| 2.2 | | | | 775 |
| 3.1 | *Florella pascuensis* | Agat, Guam | June 2008 | 78 |
| 3.2 | | | | 367 |
| 4 | *Pseudictyota dubia* | San Pedro, California | June 2009 | 438 |
| 5 | *Striatella unipunctata* | Florida Bay, Florida | March 2010 | 674 |
| 6 | *Hanicella moenia* | Gab Gab, Guam | August 2011 | 486 |
| 7 | *Paralia cf. longispina* | Gab Gab, Guam | August 2011 | 177 |
| 8 | *Astrosyne radiata* | Gab Gab, Guam | July 2013 | 283 |
| 9 | *Neosynedra provincialis* | Pickle's Reef, Florida | April 2014 | 505 |
| 10 | *Leptocylindrus danicus* | Gulf of Mexico, off Texas coast | August 2011 | 1,571 |

[a]More detailed collection data, including latitude and longitude of collection sites, can be found in Table S1.
[b]Different cultures and successive time points for a given culture are represented; e.g., samples 2.1 and 2.2 are samples from culture 2 taken at different time points.
[c]Number of days elapsed between initial diatom cultivation and pellet harvesting for DNA extraction.

generally held for the multiply sampled cultures (see Fig. S2 in the supplemental material); both culture 1 and culture 3 (*A. radiata* and *F. pascuensis*) had increased Shannon and inverse Simpson scores over time as well as decreased dominance by the most abundant OTU.

These trends in evenness diversity are easily visualized in the changing abundance of the major OTUs over time in our most highly sampled culture (Fig. 2). Culture 1, featuring the diatom *A. radiata*, was sampled four times over a period spanning over 7 years. Figure 2 depicts OTU dynamics in culture 1 over time as a Muller plot, with each colored field representing a single OTU. With a few exceptions, strains that were ever abundant were detectable at all four time points, but their relative abundance changed dramatically over time, with previously dominant groups dropping to the limit of detection by the final time point, and once nearly undetectable groups eking

**TABLE 2** Diversity metrics

| Sample | No. of OTUs observed | Coverage (%)[a] | Diversity index | | | % most abundant OTU | No. of ubiquitous OTUs[d] | % ubiquitous[e] |
|---|---|---|---|---|---|---|---|---|
| | | | Chao[b] | Shannon[c] | Inverse Simpson[b] | | | |
| 1.1 | 172 | 98.4 | 395 (293–582) | 2.44 ± 0.05 | 4.0 (3.8–4.1) | 48.2 | 31 | 13.5 |
| 1.2 | 103 | 99.0 | 246 (168–416) | 1.86 ± 0.04 | 3.6 (3.5–3.7) | 37.7 | 22 | 8.9 |
| 1.3 | 117 | 99.1 | 238 (173–379) | 2.66 ± 0.04 | 7.8 (7.6–8.1) | 22.0 | 27 | 8.9 |
| 1.4 | 196 | 98.1 | 415 (322–578) | 3.39 ± 0.04 | 17.5 (16.8–18.2) | 13.7 | 28 | 33.2 |
| 2.1 | 72 | 99.4 | 117 (90–183) | 2.15 ± 0.04 | 5 (4.8–5.2) | 36.9 | 22 | 42.1 |
| 2.2 | 89 | 99.4 | 155 (116–251) | 1.70 ± 0.05 | 2.6 (2.5–2.7) | 59.4 | 22 | 72.6 |
| 3.1 | 118 | 98.8 | 223 (173–319) | 1.79 ± 0.04 | 3.1 (3.0–3.2) | 52.4 | 19 | 55.7 |
| 3.2 | 117 | 99.1 | 253 (177–424) | 2.71 ± 0.04 | 8.7 (8.4–9.0) | 22.8 | 23 | 18.5 |
| 4 | 105 | 99.2 | 195 (144–312) | 1.95 ± 0.05 | 3.4 (3.2–3.5) | 50.3 | 24 | 55.6 |
| 5 | 144 | 99.0 | 217 (180–292) | 3.23 ± 0.04 | 13.7 (13.1–14.4) | 19.4 | 19 | 14.1 |
| 6 | 128 | 99.1 | 183 (155–241) | 2.86 ± 0.04 | 10.5 (10.1–10.9) | 18.6 | 24 | 16.5 |
| 7 | 298 | 97.2 | 678 (535–908) | 3.73 ± 0.05 | 17.7 (16.9–18.7) | 14.7 | 23 | 16.9 |
| 8 | 63 | 99.5 | 99 (78–153) | 1.54 ± 0.04 | 3.0 (2.9–3.1) | 51.5 | 18 | 79.6 |
| 9 | 92 | 99.1 | 228 (152–399) | 2.10 ± 0.04 | 5.1 (4.9–5.2) | 33.6 | 18 | 42.6 |
| 10 | 109 | 99.4 | 142 (123–186) | 2.63 ± 0.04 | 7.2 (6.9–7.4) | 26.6 | 21 | 4.8 |
| Mean | 128.2 | 98.9 | 252.3 | 2.45 | 7.52 | 33.9 | 22.7 | 32.2 |

[a]Good's coverage index (85).
[b]Chao and inverse Simpson values are estimates, with 95% confidence intervals in parentheses.
[c]Shannon values are means, and the ± value indicates the width of the 95% confidence interval.
[d]"Ubiquitous" OTUs are those found in more than half of samples.
[e]The percentage of total sequences in each sample that were from the 32 ubiquitous OTUs.

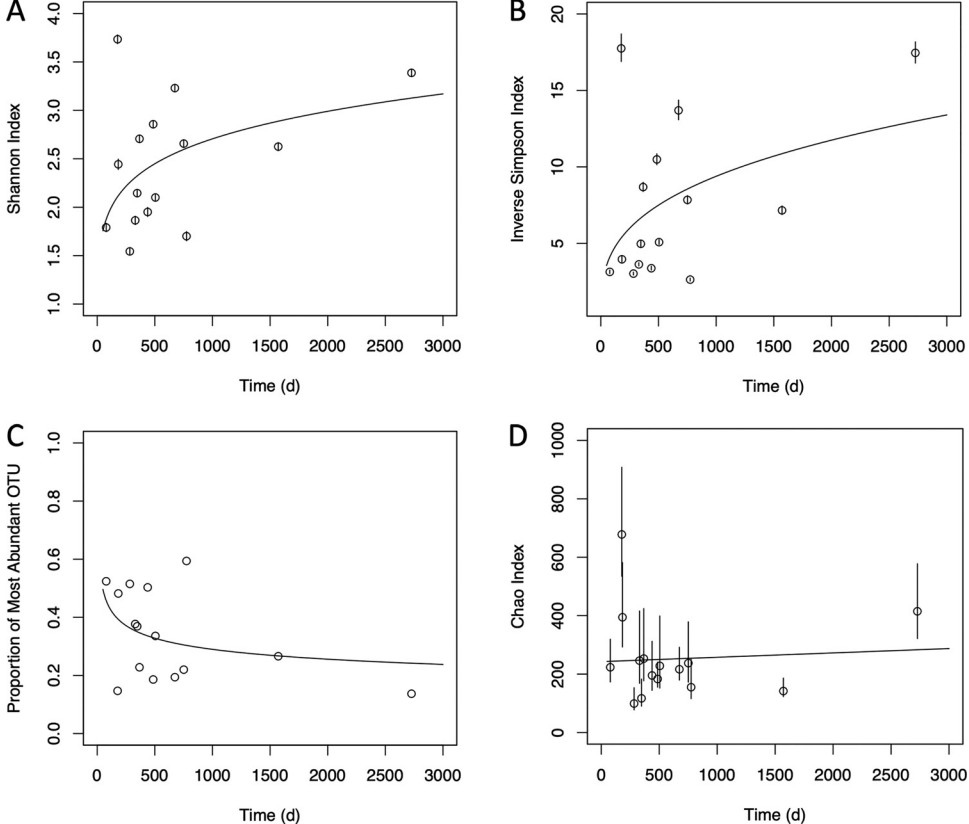

**FIG 1** Changes in culture diversity metrics over time. Points represent parameter estimates, and error bars show the 95% confidence intervals. Fit lines are derived from the best-fit model according to BIC analysis; for panels A to C, the best-fit model was a one-parameter power law, whereas for panel D, the best fit was from a simple linear model. The slope in panel D was not significantly different from zero. Model parameters fitted to data sets omitting the latest time point were not significantly different from those determined with the full data set.

out a significant portion of the community in later samplings. These dynamics suggest a complex system of ecological and evolutionary forces acting on these seemingly simple microalgal laboratory cultures while simultaneously conserving diversity over time.

We also investigated whether diatom species or collection site/time had a persistent impact on microbiome structure. We found that cultures collected at the same

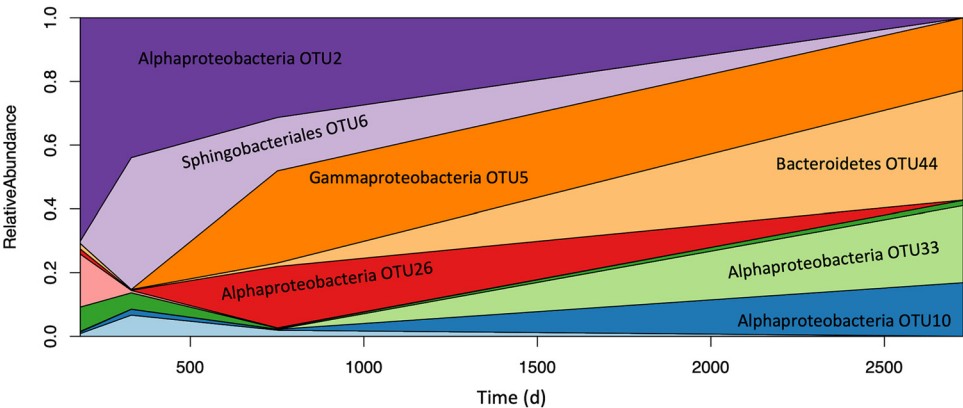

**FIG 2** Bacterial community dynamics in *Astrosyne radiata* culture 1. The culture was sampled four times over a period of approximately 8 years. Different colors represent different OTUs, with the vertical area covered by each representing its share of the community at a given time point. Note that the leftmost limit of the *x* axis represents the time of the first sample collection (183 days), not the time of culture isolation (i.e., 0 days).

location tended to have similar microbiomes years later despite having different diatom species (Table S3). Overall, there were significantly more OTUs shared between samples from the same site (mean, 39) than between samples collected from different sites (mean, 27.7) (Fig. S3A) (Mann-Whitney U test, $P = 0.007$). There were also significantly more OTUs in common between samples of cultures collected at Gab Gab Beach on the same day than were shared between cultures collected from the same site but in different years (Fig. S3B) (Mann-Whitney U test, $P = 0.002$). In contrast, there was not a significant relationship between the number of shared OTUs between two samples and whether they came from cultures of the same diatom species (Table S4 and Fig. S4A) (Mann-Whitney U test, $P = 0.22$), largely because our two independently collected *A. radiata* cultures shared relatively few OTUs, with only 9 found in common across all five samples and an average of 19.8 OTUs shared between the four culture 1 samples paired with culture 8, compared to an average of 55.8 in common between the four culture 1 samples (Fig. S4B). However, we note that we based this observation on only two independently collected *A. radiata* cultures; greater sampling might have revealed an effect of diatom species not detectable in our data set.

The primacy of collection site/time over taxonomy in determining microbiome structure is supported by analysis of molecular variance (AMOVA), where all four distance matrices we considered indicated overall significant differences ($P < 0.05$) for site/time, but none for either diatom species or order (Tables S5 and S6). After correcting for multiple comparisons during *post hoc* pairwise contrasts of sites, there were marginal but nonsignificant differences between the 2008 and 2011 Gab Gab samplings and between Gab Gab 2008 and Achang Reef (Table S7). Differences were driven by a relatively small number of OTUs (Tables S8 and S9). OTU2, an unclassified member of the *Alphaproteobacteria*, was a major driver, as it was initially highly abundant in the *A. radiata* culture obtained from Gab Gab in 2008 but was absent from the other sites. While less abundant overall, OTU57, identified as *Balneola* in the phylum *Bacteroidetes*, was also important for differentiating the three sites.

In addition to these apparently site-specific taxa, we also observed several OTUs that were found in most cultures regardless of species or collection site (Table 3; Fig. S5). Nineteen OTUs were present in at least 2/3 of the samples, including the most abundant OTU, identified as a strain of the prosthecate bacterium *Hyphomonas*. An additional 13 OTUs were found in at least half of the samples. Of these 32 ubiquitous strains, 12 were affiliated with the *Alphaproteobacteria*, including the 6 most abundant overall, and all but three of the remainder were from either the *Gammaproteobacteria*, *Bacteroidetes*, or *Firmicutes*. The abundances of certain of these ubiquitous taxa were correlated with each other, and five clusters emerged from our hierarchical clustering analysis, with three (clusters 1, 2, and 4 in Fig. 3) presenting significantly positive correlations between at least 90% of pairwise comparisons within the cluster. The first and largest cluster contained 8 OTUs and was dominated by two alphaproteobacterial OTUs (*Labrenzia* and an unclassified OTU). A second, smaller cluster contained only 3 alphaproteobacterial OTUs, *Oceanicola*, *Tistlia*, and an unidentified member of the *Rhodobacteraceae*. Cluster 4 contained relatively low-abundance OTUs and was dominated by an unclassified member of the *Enterobacteriaceae*. Some of the low-abundance OTUs, including the *Enterobacteriaceae* OTU dominating cluster 4, are members of taxa that have been implicated as contaminants present in PCR and sequencing reagents (36), suggesting that the appearance of their ubiquity was an artifact. However, the first two clusters are dominated by marine organisms, and their intercorrelation perhaps suggests mutually reinforcing interactions within those groups.

The stability and lack of species specificity of our diatom microbiomes is also visible in Fig. 4, which positions each sample as the result of a 3-dimensional nonmetric dimensional scaling (NMDS) ordination based on the abundance-informed Jaccard distance. The four ovals each encompass samples collected at the same time from the same location, and arrow vectors connect older samples to more recent ones. It is clear from this plot that the microbiomes are dynamic, yet it is also clear that 3 of the 4 ovals

**TABLE 3** Ubiquitous OTUs[a]

| OTU | Phylum or class | Genus | No. of samples[b] | % of sequences[c] |
|-----|-----------------|-------|-------------------|-------------------|
| OTU1 | *Alphaproteobacteria* | *Hyphomonas* | 14 | 8.62 |
| OTU3 | *Alphaproteobacteria* | *Sphingomonadales*, unclassified | 8 | 3.65 |
| OTU4 | *Alphaproteobacteria* | *Oceanicola* | 13 | 4.47 |
| OTU8 | *Alphaproteobacteria* | *Roseibium* | 9 | 2.19 |
| OTU10 | *Alphaproteobacteria* | *Alphaproteobacteria*, unclassified | 12 | 2.09 |
| OTU18 | *Alphaproteobacteria* | *Rhodobacteraceae*, unclassified | 9 | 1.35 |
| OTU19 | *Gammaproteobacteria* | *Gammaproteobacteria*, unclassified | 9 | 1.55 |
| OTU28 | *Bacteroidetes* | *Balneola* | 13 | 0.71 |
| OTU31 | *Alphaproteobacteria* | *Labrenzia* | 12 | 0.75 |
| OTU33 | *Alphaproteobacteria* | *Alphaproteobacteria*, unclassified | 12 | 1.23 |
| OTU34 | *Bacteroidetes* | *Ekhidna* | 14 | 3.83 |
| OTU64 | *Gammaproteobacteria* | *Enterobacteriaceae*, unclassified | 15 | 0.28 |
| OTU73 | *Gammaproteobacteria* | *Gammaproteobacteria*, unclassified | 12 | 0.18 |
| OTU77 | *Alphaproteobacteria* | *Tistlia* | 8 | 0.17 |
| OTU100 | *Bacteroidetes* | *Flammeovirgaceae*, unclassified | 11 | 0.23 |
| OTU101 | *Alphaproteobacteria* | *Alphaproteobacteria*, unclassified | 8 | 0.14 |
| OTU107 | *Bacteroidetes* | *Flammeovirgaceae*, unclassified | 8 | 0.15 |
| OTU129 | *Alphaproteobacteria* | *Rhodospirillaceae*, unclassified | 9 | 0.07 |
| OTU136 | *Firmicutes* | *Veillonella* | 14 | 0.08 |
| OTU140 | *Fusobacteria* | *Fusobacterium* | 11 | 0.09 |
| OTU152 | *Bacteroidetes* | *Prevotella* | 12 | 0.07 |
| OTU196 | *Firmicutes* | *Streptococcus* | 12 | 0.03 |
| OTU197 | *Actinobacteria* | *Atopobium* | 10 | 0.04 |
| OTU198 | *Alphaproteobacteria* | *Rickettsia* | 15 | 0.08 |
| OTU202 | *Gammaproteobacteria* | *Pasteurellaceae*, unclassified | 8 | 0.03 |
| OTU227 | *Firmicutes* | *Streptococcus* | 10 | 0.03 |
| OTU229 | *Firmicutes* | *Gemella* | 10 | 0.03 |
| OTU246 | *Firmicutes* | *Lactobacillus* | 10 | 0.02 |
| OTU255 | *Firmicutes* | *Staphylococcus* | 8 | 0.02 |
| OTU258 | *Firmicutes* | *Streptococcus* | 8 | 0.01 |
| OTU307 | *Bacteroidetes* | *Prevotella* | 8 | 0.01 |
| OTU320 | *Actinobacteria* | *Mycobacterium* | 9 | 0.01 |

[a]OTUs were deemed ubiquitous if they were observed in at least half of all 15 samples.
[b]Number of samples (of 15 total) in which the OTU was detected.
[c]Percentage of total sequences across all 15 samples contained within the OTU.

encompass points that remain clustered together even after long periods of time—the four points from culture 1, for instance, span 7 years. On the other hand, cultures of the same species from different sampling years are not closely clustered at all; the purple sphere at the far left of Fig. 4 represents culture 8, which is also *A. radiata* from Gab Gab Beach but collected 5 years after culture 1 and is as far from the points of culture 1 as any other sample in the plot. Figure S6 depicts the same plot as an animation with vectors showing the most influential OTUs for determining a sample's position. Only one of these OTUs, *Hyphomonas* OTU1, was also ubiquitous; the remainder were relatively specific to certain cultures (e.g., OTU2, which we have already seen was a major defining taxon separating culture 1 from others).

Finally, several relatively abundant taxa were unclassifiable below the domain level, suggesting that these cultures harbored previously uncultured, novel bacterial taxa. Among the OTUs that were either found in >50% of cultures or represented at least 5% of any one culture, we found nine OTUs classified only as *Bacteria* (Table 4). From these, the two most abundant were most closely affiliated (~85% sequence identity) with *Leptospira* or *Leptonema*, genera in the phylum *Spirochaetes*. Three more were most closely associated (87% to 98% sequence identity) with the phylum *Planctomycetes*, and another with the family *Rhodobiaceae*. A final three relatively uncommon OTUs were most closely related to diatom mitochondria and likely escaped our quality control procedures due to the paucity of diatom mitochondrial sequences in the ARB-SILVA database.

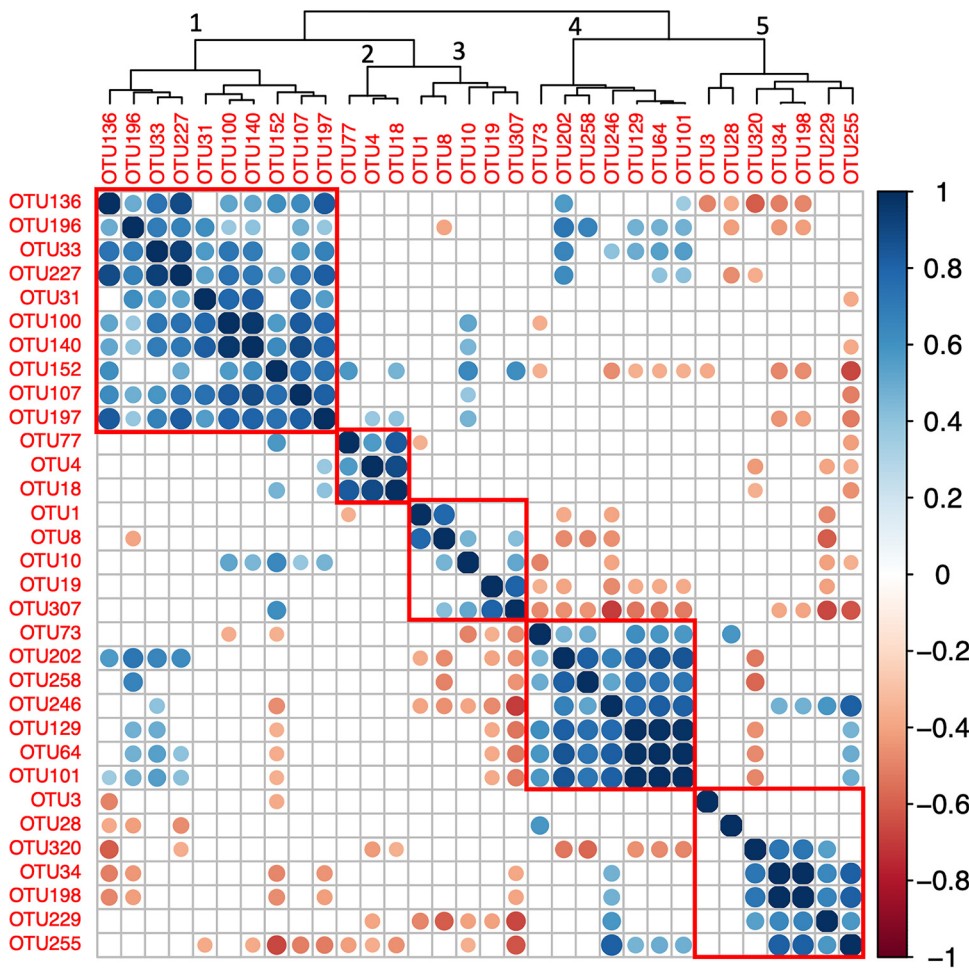

**FIG 3** Correlation between ubiquitous OTUs. The colors and sizes of circles indicate the Pearson correlation coefficient between the abundances of each pair of ubiquitous (present in >50% of samples) OTUs across samples. Empty boxes indicate correlations that were not large enough to be statistically significant ($P > 0.05$). OTUs were arranged by hierarchical clustering; red rectangles outline the five modules selected by the clustering algorithm. Numbers above nodes in the clustering tree correspond to the numbering of the modules as discussed in the text.

## DISCUSSION

It is important to acknowledge that the cultures we studied were not collected with the intention of testing hypotheses about diatom microbiome stability over time. The DNA pellets were collected for other purposes, and we exploited them as a preliminary investigation into the impact of long-term cultivation on diatom microbiome structure. As such, our conclusions are not as robust as they would be in a planned and well-replicated experiment. For instance, because we do not have sequences of the ambient community from the original sample sites taken at the time of cultivation or samples taken during the early weeks of culture maintenance, we cannot draw robust conclusions about the similarity or lack thereof between our cultures and the natural communities from which the diatoms came; indeed, it is known that phycosphere communities change rapidly in the early months after cultivation (37). We also acknowledge that we cannot be certain that none of the bacteria we detected were contaminants, either introduced over the years during laboratory handling or as contaminants of our DNA extraction or sequencing reagents. Nevertheless, our results appear to be consistent with previous reports of remarkably diverse microalgal phycosphere-bacterial communities that are similar under laboratory and culture-independent conditions (38, 39). We hope that future research will

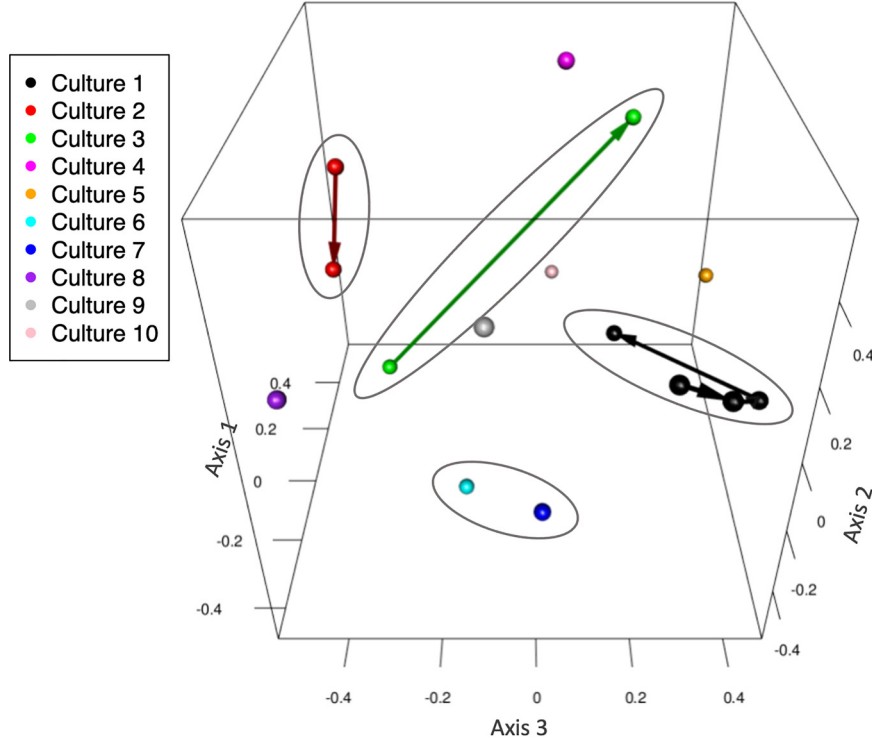

**FIG 4** Diatom microbiome community structure. This NMDS plot shows how the various cultures cluster in three-dimensional space. Arrows connect multiply sampled cultures, going from earlier to later samples. Samples from cultures sharing a common sampling origin are circled. An animated version of this plot that also contains vectors showing the impact of key OTUs on sample placement is available as Fig. S6.

approach these issues more systematically, although we note the difficulty inherent in maintaining experiments over the time scales represented by some of the cultures studied here and of completely avoiding introduction of laboratory contaminants during years of continuous culture maintenance.

Despite the above constraints, we may confidently derive three main conclusions from our results. First, and unexpected based on our initial assumptions, OTU evenness diversity increased over time in our cultures. Second, the diatom-associated microbiomes were differentiated based on the local conditions at the time of diatom isolation rather than the species identity of the cultured diatom. Third, despite differences between sampling sites, a subset of OTUs was ubiquitous across many cultures, and in some cases these ubiquitous taxa appeared to reinforce each other as intercorrelated clusters. Although empirical evidence for specific mechanisms is lacking due to the limitations

**TABLE 4** OTUs representing novel bacterial taxa

| OTU | No. of cultures | Abundance (%) | | Best hit[a] | % identity[a] |
|-----|-----|-----|-----|-----|-----|
| | | Total | Maximum | | |
| OTU7 | 2 | 2.2 | 19.4 | *Leptospira/Leptonema* | 84.86 |
| OTU14 | 4 | 1.4 | 20.9 | *Leptospira/Leptonema* | 85.77 |
| OTU15 | 4 | 1.1 | 15.1 | *Planctomycetes* | 86.51 |
| OTU20 | 1 | 0.9 | 12.9 | *Planctomycetes* | 94.07 |
| OTU46 | 1 | 0.4 | 6.5 | *Rhodobiaceae* | 84.92 |
| OTU50 | 1 | 0.5 | 8.2 | *Planctomycetes* | 98.02 |
| OTU51 | 2 | 0.4 | 6.1 | *Eucampia zodiacus* mitochondrion | 82.28 |
| OTU61 | 4 | 0.6 | 6.1 | *Psammoneis japonica* mitochondrion | 84.58 |
| OTU127 | 1 | 0.8 | 11.4 | *Didymosphenia geminata* mitochondrion | 87.7 |

[a]Closest match to the representative OTU sequence from a BLASTn search.

of our methodology, we believe that these observations suggest three structuring forces acting on these diatom microbiomes: pervasive negative frequency dependence, historical contingency, and selection for a globally present diatom microbiome. We address each of these forces in turn below.

**Negative frequency dependence.** The large number of taxa present in each culture, even after years of growth in culture, was surprising to us. We observed that time since cultivation played a key role in determining microbiome structure, but with the opposite effect from what we expected. We had anticipated that young cultures would have high diversity, reflecting the conditions typically present in natural seawater, but that over time the large majority of taxa would be competitively excluded due to the constancy and relative simplicity of the laboratory environment, as well as the contrast between the culture medium and the natural marine environment. Instead, we observed the opposite trend, with evenness diversity increasing with time (Fig. 1). Several eco-evolutionary mechanisms could underpin this observation. First, the community could be composed of many taxa with similar nutritional requirements but also similar affinities for the limiting factor, such that the rate of competitive exclusion is low relative to the rate of adaptive evolution. This scenario is similar to the phenomenon of clonal interference observed in the evolution of asexual populations (40) and could potentially preserve a high degree of species richness indefinitely. However, it is difficult to reconcile this mechanism with the observation of increased species evenness over time, since the more numerous species would always have a statistical advantage in terms of acquiring incremental beneficial mutations and would therefore be unlikely to become less dominant over time.

Another possible explanation for the observed trend in evenness diversity is that many of the strains present have higher fitness when rare, preventing their competitive exclusion. This phenomenon of negative frequency-dependent selection has been observed in many systems and is a key way that diversity can be preserved among competitors living in sympatry (41, 42). The black queen hypothesis describes a process by which negative frequency dependence readily arises in microbial communities as natural selection favors strains that streamline their metabolisms by evolving dependency on other community members for costly "leaky" products and functions (43, 44). Essentially, the release of compounds into the extracellular medium by members of the diatom-associated consortium, either purposely or as a product of cell lysis, would seed an unexpectedly complex "marketplace" that could provide many opportunities for diversification and specialization both before and after cultivation (45). Importantly, negative frequency dependence can slow competitive exclusion and stabilize coexistence of competing lineages even on time scales of thousands of generations (46). Moreover, black queen interactions tend to favor coexistence at relatively stable equilibrium frequencies (47–49), which would be consistent with the evening-out of the communities over time that we observed as well as the reproducible clustering among the ubiquitous taxa (Fig. 3).

**Historical contingency.** The fact that the numerically dominant OTUs in each sample as well as the OTUs that were most important in positioning samples in coordinate space were relatively infrequently present across samples suggests that long-term community structure in these cultures was highly dependent on stochastic initial conditions. For example, the alphaproteobacterium OTU2 was the most abundant OTU present in culture 1 for almost 2 years but was mostly absent from all other samples, including other cultures from Gab Gab Beach taken on subsequent dates. The simplest explanation for this observation is that OTU2 was, for whatever reason, a strong competitor under laboratory conditions but rare in the environment, such that most cultures would not contain even a single OTU2 cell when they were initiated. The strong dependency of community composition on early, random colonization events, sometimes called the lottery effect, has been shown to be important in the formation of coral reef assemblages as well as the microbiota of macroalgal thalli (50, 51), and such an effect also seems to be a major factor in culture microbiome assembly. Based on other studies showing that microbiome composition can have a powerful positive or negative impact on growth rate and other microalgal phenotypes (13, 19, 52, 53), this

historical contingency may have an important effect on our interpretation of microalgal physiology in culture-based experiments.

**The core diatom microbiome.** In contrast to the rarely present yet influential OTUs described above, another group of taxa were widely present despite cultures being collected at widely dispersed locations and times. While we did not sequence the ambient microbiome of each culture collection site, the diatom microbiomes had substantially different structures than is typical for coastal waters (54), in particular with the absence of cyanobacteria and SAR11. It is possible that some of the ubiquitous taxa, and in particular those with lower abundance, are contaminants from the various reagents used to extract, process, and sequence community DNA (36). However, many of the ubiquitous strains are common in marine ecosystems and have been observed in other diatom microbiomes (6). The observation that such closely related organisms are present from Guam to the Gulf of Mexico in numbers large enough to be randomly sampled is striking and supportive of the "everything is everywhere" maxim (55). But the other half of Baas-Becking's quote, "but the environment selects," is even more evident, as the most likely explanation for the frequent discovery of these taxa in our cultures is that the diatom culture environment selected for them despite low environmental abundance. In contrast to OTUs whose infrequent presence may have reflected stochastic lottery effects, the presence of these ubiquitous OTUs would seem to reflect natural selection by the diatom and potentially provides insight into *in situ* phycosphere processes (37–39, 52, 56–58).

Most of our attention so far has focused on broad trends in community structure, but some ubiquitous OTUs deserve special attention. The most abundant OTU across all our samples (14/15 samples, 8.6% of total sequences) was *Hyphomonas*, a *Caulobacter*-like genus of stalked, asymmetrically dividing bacteria with a complex life cycle involving alternating motile and permanently attached life cycle stages (59). *Hyphomonas* has been observed previously in association with diatoms in nature by scanning electron microscopy (SEM) imaging and DNA sequencing (60) and has also been observed to be a prominent component of the microbiome of other pennate diatoms (7, 61, 62) and macroalgae (50, 63). The sessile, stalked morphotype of *Hyphomonas* produces robust biofilms and has been implicated as a pioneer organism in the formation of marine biofilms and a nucleator of biofouling (59). Indeed, its life history strategy is well suited to coexistence with larger microalgae, which could form a substrate for its attached life cycle stage. Other ubiquitous taxa, including *Labrenzia*, *Roseibium*, and *Tistlia*, come from groups known to contain strains that express phototrophy or nitrogen fixation genes (64–67). If present, these metabolisms could create separate niches for these organisms, as well as increasing the diversity of metabolites available in the extracellular medium, helping to explain the high evenness diversity observed in our cultures. Finally, our cultures contained several OTUs that were unclassifiable using our pipeline but whose closest relatives fall within groups that are traditionally difficult to culture. Two *Spirochaetes*-related OTUs were quite abundant, representing about 4% of total sequences and around 20% of sequences in their respective cultures. Three OTUs related to the *Planctomycetes* were also abundant, representing 2.5% of total sequences and 8 to 15% of their culture. *Planctomycetes* in particular is a group that is poorly represented in cultures, despite its importance in many ecosystems and its dominance in the important biogeochemical process of anaerobic ammonia oxidation (68). Like *Hyphomonas*, planctomycete bacteria produce a specialized holdfast and alternate between swimming and sessile life stages, making them ideal symbionts for larger microalgae. Some planctomycetes can also form robust biofilms during diatom bloom conditions (69, 70), and others produce secondary metabolites with antibiotic and algicidal activity (71). Collectively, these observations further support the hypothesis that the culture environment is metabolically complex and suggest that microalgal cultures and perhaps filtered microalgal exudates may be fruitful tools for cultivating the "unculturable" majority of marine bacteria.

In conclusion, our results support previous studies (52, 56, 72, 73) showing that microalgal microbiomes remain relatively stable over time in culture but extend those findings to a substantially longer time frame. Moreover, the inclusion of more diatom

taxa and worldwide collection sites suggests the outlines of a core diatom microbiome. Researchers have long observed that the behavior of axenic microalgal cultures is unrepresentative of natural behavior, but the presence of uncontrolled bacterial assemblages creates the possibility of unacceptably high interculture and interexperimental variability in culture-based studies. An improved empirical understanding of the "preferred" microbiome of diatoms may allow researchers to use defined consortia in experiments in order to more closely approximate the *in situ* conditions of these organisms. Also, the development of robust, stable bacterium-diatom consortia may help protect industrial microalgal cultures from infection and may increase their growth rates and/or biomass yields. Future efforts should seek to develop in more detail an understanding of the mechanisms underpinning coexistence in these consortia, using a combination of culture-based, molecular, and modeling approaches.

## MATERIALS AND METHODS

**Culture maintenance.** Cells were isolated from collections made primarily with a plankton net with 20-$\mu$m mesh or by collection of the top millimeters of sediment from the benthos (Table S1). Individual cells were identified in a Zeiss Axiovert 25 inverted microscope and isolated with a glass Pasteur pipet (35) into 15- by 100-mm glass test tubes in approximately 12 ml of liquid F/2 medium with $1.06 \times 10^{-4}$ M Na$_2$SiO$_3$ · 9H$_2$O (35, 74) at a salinity of 32 to 35 ppt. The F/2 base was seawater collected from the Texas coast of the Gulf of Mexico and passed through a 0.22-$\mu$m filter. Isolates were maintained under natural light from a north-facing window between 20 and 24°C or, in the case of the isolates from Guam, in a Percival growth chamber on a 12:12 light-dark cycle at 27°C under 30 $\mu$mol photons m$^{-2}$ s$^{-1}$ of illumination. Once microalgal growth was confirmed by microscopy, strains were maintained in triplicate with cells (approximately 0.25 ml) from one tube transferred to three fresh medium tubes every 3 to 4 months, and the remaining tubes were harvested (see below). Strain identity was confirmed by light and scanning electron microscopy of the cleaned frustules. References used to identify the diatom strains are provided in Table S1.

**DNA isolation.** During the strain transfer cycle, one replicate tube was harvested with a Pasteur pipet into a 1.5-ml microcentrifuge tube and centrifuged in an Eppendorf 5414 C microcentrifuge at 8,000 rpm for 10 min. Liquid medium was decanted off the pellet, and the pellet was stored at −80°C until DNA extraction. DNA was extracted from the pellets using a MoBio Powersoil DNA kit involving a bead-beating lysis step, following the manufacturer's protocol. All pellets were collected from diatom cells in the stationary phase of their batch culture growth cycle. No effort was made to quantify bacteria remaining in the supernatant after centrifugation.

**DNA sequencing and curation.** DNA sequencing was performed at the UAB Genomics Core Lab. The V4 region of the bacterial 16S rRNA gene was amplified using bar-coded PCR primers (F515, 5'-CACGGTCGKCGGCGCCATT-3', and R806, 5'-GGACTACHVGGGTWTCTAAT-3') (75), purified by gel electrophoresis, and then sequenced using the Illumina MiSeq platform, yielding 2,245,770 250-bp paired-end reads across all samples. Of the 17 samples originally considered, 2 had very low coverage and were removed from further consideration; the remaining samples are detailed in Table 1. DNA sequences were assembled and quality-controlled in mothur (v1.44.3) according to the MiSeq standard operating procedure (SOP) (76, 77), using vsearch (v2.13.3) to identify and remove chimeric sequences (78). After quality control and removal of identifiable chloroplast or mitochondrial sequences, 827,446 high-quality sequences remained, with individual samples ranging from 5,524 to 96,216 sequences with a median value of 61,784 sequences. These sequences were assigned to OTUs at the 3% difference threshold and were classified by reference to the SILVA small-subunit-rRNA database, version 128 (79).

**Sequence analysis and statistics.** Curated sequences were subsampled to the level of the least deeply sequenced sample (5,524 sequences) with a ranked subsampling algorithm (80). Based on Good's coverage analysis in mothur, this depth of sequencing granted >97% coverage of all samples (Table 2); rarefaction analysis also supports the conclusion that the samples were well covered (Fig. S1). Alpha diversity was quantified in mothur with the Chao richness index as well as the Shannon and inverse Simpson evenness indices. All beta diversity analyses were performed on distance matrices generated using either the theta-YC, Bray-Curtis, simple Jaccard, or abundance-informed Jaccard metric, executed in mothur. These metrics were chosen to provide a wide sampling of distance calculation strategies, including both differential abundance and presence/absence focused metrics. AMOVA comparisons (81) of multiply sampled cultures, diatom taxonomic groupings, and collection sites/dates were performed on all four matrices, and significance levels are presented as the consensus of all four. OTUs underlying significant AMOVA results were identified using Metastats (82). Ordination was performed using either principal coordinates or nonmetric dimensional scaling (NMDS) analysis on all four distance matrices, and the analysis method and distance matrix that yielded the highest $r^2$ value on 3 axes (NMDS on Jaccard abundance matrix; $r^2 = 0.69$, stress = 0.17) was used for subsequent analysis. Linear regression, curve-fitting, and data visualization were performed in R (version 4.0.3). Significance of trends in fit curves was assessed if the curve fit parameter was significantly greater than 0. Hierarchical clustering was performed on a matrix of correlation coefficients of pairs of ubiquitous taxa using the Ward algorithm and the optimum number of clusters for analysis was determined by the majority rule of 30 different methods using the NbClust package in R (83). Closest matches to OTUs unidentified below the domain level were obtained using BLASTn (84) via the NCBI

web interface with standard settings, excluding models and uncultured/environmental samples from the results.

**Data availability.** Sequence data are available at the NCBI SRA Archive under BioProject no. PRJNA706454. Raw data and code for sequence and statistical analysis are included with the online supplemental material and are also deposited at BCO-DMO: https://www.bco-dmo.org/project/752763.

## SUPPLEMENTAL MATERIAL

Supplemental material is available online only.

**SUPPLEMENTAL FILE 1**, GIF file, 19.4 MB.
**SUPPLEMENTAL FILE 2**, PDF file, 1.2 MB.
**SUPPLEMENTAL FILE 3**, CSV file, 0.1 MB.
**SUPPLEMENTAL FILE 4**, CSV file, 0.1 MB.

## ACKNOWLEDGMENTS

We are grateful to Casey Morrow and the UAB Genomics Core Lab for assistance with 16S rRNA sequencing and to Schonna Manning and Edward Theriot at UT Austin for logistical support and culture maintenance.

This work was partially supported by a grant from the National Science Foundation (OCE-1851085) and an Early Career Fellowship from the Simons Foundation to J.J.M., as well as a National Science Foundation Graduate Research Fellowship (DGE-1945997) to M.W. The funders had no role in study design, data collection and interpretation, or the decision to submit the work for publication. Any opinions, findings, and conclusions or recommendations expressed in this material are those of the authors and do not necessarily reflect the views of the National Science Foundation.

We declare no conflicts of interest.

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
