## [Reviewer comments · Microbiology Spectrum]

**Microbiology
Spectrum**

STRUCTURE AND LONG-TERM STABILITY OF THE MICROBIOME IN DIVERSE DIATOM CULTURES

Marcelo Barreto Filho, Melissa Walker, Matt Ashworth, and J. Jeffrey Morris

Corresponding Author(s): J. Jeffrey Morris, University of Alabama at Birmingham

Review Timeline:

Submission Date:	May 24, 2021
Editorial Decision:	May 26, 2021
Revision Received:	May 28, 2021
Accepted:	May 28, 2021

Editor: Jeffrey Gralnick

Reviewer(s): The reviewers have opted to remain anonymous.

Transaction Report:

DOI: <https://doi.org/10.1128/Spectrum.00269-21>

May 26, 2021

Dr. J. Jeffrey Morris
University of Alabama at Birmingham
Department of Biology
Campbell Hall 253
Birmingham, Alabama 35294-1170

Re: Spectrum00269-21 (STRUCTURE AND LONG-TERM STABILITY OF THE MICROBIOME IN DIVERSE DIATOM CULTURES)

Dear Dr. J. Jeffrey Morris:

Thank you for submitting your manuscript to Microbiology Spectrum. You and your co-authors have done a nice job thoughtfully responding to the reviewers at AEM. In consultation with another editor, it was suggested that some additional discussion be added (maybe in part 1 of the discussion that acknowledges limitations, particularly regarding absence of samples in early stages of cultivation that reviewers commented on) with some supporting refs on what is known about how phycosphere communities rapidly changes upon cultivation in the lab, for example:

<https://aem.asm.org/content/73/9/3117.short>

<https://www.nature.com/articles/s41396-020-00812-x>

I am returning your manuscript with 'minor modifications' so you can consider the above point. If you agree that this adds to the discussion, please upload a revised manuscript into the system and re-submit (you shouldn't need to change any other files). If there is a reason not to include this in the discussion, please let me know in a revised cover letter. I am looking forward to seeing this published in Spectrum.

Please use this link to submit your revised manuscript - we strongly recommend that you submit your paper within the next 60 days or reach out to me. Detailed information on submitting your revised paper are below.

Link Not Available

Sincerely,

Jeffrey Gralnick

Journals Department
Staff Comments:

Preparing Revision Guidelines

For complete guidelines on revision requirements, please see the Instructions to Authors at [link to page]. **Submissions of a paper that does not conform to Microbiology Spectrum guidelines will delay acceptance of your manuscript.**

Please return the manuscript within 60 days; if you cannot complete the modification within this time period, please contact me. If you do not wish to modify the manuscript and prefer to submit it to another journal, please notify me of your decision immediately so that the manuscript may be formally withdrawn from consideration by Microbiology Spectrum.

If you would like to submit an image for consideration as the Featured Image for an issue, please contact Spectrum staff.

May 28, 2021

Dr. J. Jeffrey Morris
University of Alabama at Birmingham
Department of Biology
Campbell Hall 253
Birmingham, Alabama 35294-1170

Re: Spectrum00269-21R1 (STRUCTURE AND LONG-TERM STABILITY OF THE MICROBIOME IN DIVERSE DIATOM CULTURES)

Dear Dr. J. Jeffrey Morris:

Your manuscript has been accepted, and I am forwarding it to the ASM Journals Department for publication. You will be notified when your proofs are ready to be viewed.

Sincerely,

Jeffrey Gralnick
Editor, Microbiology Spectrum

Supplemental Figure S6: Accept
Supplemental Table S6: Accept
Supplemental Table S1: Accept